# In vivo pharmacokinetic enhancement of monomeric Fc and monovalent bispecific designs through structural guidance

Lu Shan [1,6✉], Nydia Van Dyk [1], Nantaporn Haskins[1], Kimberly M. Cook[2], Kim L. Rosenthal[1], Ronit Mazor[1,7], Sonia Dragulin-Otto[3], Yu Jiang[4], Herren Wu[1], William F. Dall'Acqua[1], Martin J. Borrok[5], Melissa M. Damschroder[1] & Vaheh Oganesyan [1✉]

In a biologic therapeutic landscape that requires versatility in targeting specificity, valency and half-life modulation, the monomeric Fc fusion platform holds exciting potential for the creation of a class of monovalent protein therapeutics that includes fusion proteins and bispecific targeting molecules. Here we report a structure-guided approach to engineer monomeric Fc molecules to adapt multiple versions of half-life extension modifications. Co-crystal structures of these monomeric Fc variants with Fc neonatal receptor (FcRn) shed light into the binding interactions that could serve as a guide for engineering the half-life of antibody Fc fragments. These engineered monomeric Fc molecules also enabled the generation of a novel monovalent bispecific molecular design, which translated the FcRn binding enhancement to improvement of in vivo serum half-life.

[1] Antibody Discovery and Protein Engineering, R&D, AstraZeneca, Gaithersburg, MD, USA. [2] Early Oncology, R&D, AstraZeneca, Gaithersburg, MD, USA. [3] Dosage Form Design and Development, R&D, AstraZeneca, Gaithersburg, MD, USA. [4] Clinical Pharmacology and Safety Science, R&D, AstraZeneca, Gaithersburg, MD, USA. [5] Janssen BioTherapeutics, Janssen Research and Development, Spring House, PA, USA. [6] Present address: Denali Therapeutics, South San Francisco, CA, USA. [7] Present address: Division of Cellular and Gene Therapies, FDA Center for Biologics Evaluation and Research, Silver Spring, MD, USA. ✉email: shan@dnli.com; vaheh.oganesyan@AstraZeneca.com

The field of drug discovery has made remarkable progress in delivering paradigm-shifting antibody and antibody-derivative therapeutics along with breakthroughs in both disease biology and antibody technology[1,2]. The synergy between research advances in immuno-oncology and the development of novel bispecific and multispecific targeting platforms has produced many promising possibilities for driving the latest success in cancer treatment[3–7]. The importance of minimizing toxicity while achieving clinical efficacy has highlighted the need to fine-tune targeting valency from the default bivalent format of immunoglobulin Gs (IgGs) or fragment crystallizable (Fc) fusion proteins. Increasing evidence indicates that bispecific and multi-specific formats with monovalent targeting arms are necessary for reducing nonspecific cell killing, cytokine release, and undesired receptor crosslinking and improving receptor agonism and transport[8–11]. T-cell engagers or natural-killer (NK) cell engagers, such as bispecific T-cell engagers (BiTe), dual-affinity re-targeting proteins (DART), bispecific killer-cell engagers (BiKE), and tris-pecific killer-cell engagers (TriKE), have shown clinical promise with monovalent bispecific and multispecific targeting but suffer from short half-lives. Heterodimeric Fc engineering has made it possible to extend the half-lives of these engagers via the development of technologies such as knob-in-hole, CrossMab, and DuetMab, which can direct correct chain pairing for bispecific and multispecific targeting[11–13].

The successful generation of a conformationally stable, monomeric Fc antibody fragment with a "tunable" serum half-life could open new possibilities for antibody and Fc fusion therapeutics. Of the more than 180 therapeutic proteins that have received approval from the U.S. Food and Drug Administration, many have the potential to be fitted into active, longer-lasting fusion proteins[14–18]. Immune-cell engagers, antibody-drug conjugates, immunocytokine fusions, and other therapeutic proteins could be tailor-designed to provide monovalent targeting in both monospecific and multispecific formats for enhanced activity and reduced toxicity. Past efforts to engineer a monomeric Fc, defined as one set of CH2 and CH3 domains, have proved challenging due to the extensive interactions to be disrupted in the CH3–CH3 dimer interface. At high concentrations, monomer–dimer equilibrium has been observed in many mutants[19,20]. Among the engineered monomeric Fc modalities, only two molecules have been reported to have crystal structures with demonstrable homogeneity and stability[13,21]. One of these is a monomer that is stabilized by the addition of a glycosylation site that blocks the CH3–CH3 interaction[21]. The other, which was generated by our group, is a monomeric Fc derived from an IgG4 phage library that was rationally designed on the basis of previous findings[13].

To broaden the application of the monomeric Fc platform for the next wave of protein therapeutics, we set out to address three important, interconnected aspects of this endeavor: (1) a tunable serum half-life, (2) versatile construction of monovalent bispecific molecules, and (3) facile structural interrogation of Fc neonatal receptor (FcRn) interaction with Fc variants. The pH-dependent Fc–FcRn interaction is a key contributor to the prolonged serum half-life of antibodies and their derivatives. FcRn harnesses antibody molecules and carries them through the acidic endo-somal vesicles, protects them from lysosomal degradation, and releases them outside the cells due to weak binding at neutral pH[22,23]. Monomeric Fc variants are expected and observed to have reduced apparent binding to FcRn with loss of dimer avidity[13,21]. Previously, we circumvented the loss of FcRn binding by building the YTE (M252Y/S254T/T256E) mutations into the phage library template design. The resulting monomeric Fc molecule yielded greater than 10-fold improvement in FcRn binding affinity compared with its counterpart without YTE[13,24].

Aside from YTE mutations, other half-life extension mutants have been exploited to enhance the acidic binding affinity of the Fc domain while maintaining weak binding at neutral pH[24–26]. The difficulty of structurally resolving the Fc–FcRn interaction is well known to be due to the dimeric nature of Fc. It has been reported that the "growth of well-ordered [Fc–FcRn] co-crystals is apparently prevented by the packing, in which Fc homodimers bridge between dimers of FcRn heterodimers to create an 'oli-gomeric ribbon.'"[27] To obtain a different packing arrangement, investigators in one study engineered a heterodimeric Fc that contains only a single FcRn binding site, producing a 2.8 Å crystal structure of a rat Fc–FcRn complex[28]. Using a similar rationale, we took an alternative approach by adding another FcRn-binding protein, human serum albumin, to the structural complex to block "oligomeric ribbon" formation[29].

With the development of monomeric Fc molecules, there will be no need for such strenuous approaches. Monomeric Fc molecules would provide a quick and robust method for achieving structural insight at the Fc–FcRn interface. In addition, the quaternary structure of crystal packing may provide valuable information on close-proximity molecular interactions at high concentrations in solution. Therefore, we hypothesized that with an effective protein engineering approach powered by structural guidance, previously identified interface mutations could be adopted to generate monomeric Fc molecules with other half-life extension mutations.

In this work, we report a structure-guided approach to engineer a monomeric Fc molecule that is adaptable for half-life extension modifications beyond those achieved with the previously built-in YTE mutations. This is the first proof-of-concept monomeric bispecific molecular design demonstrating that it is possible to achieve a marked improvement in in vivo serum half-life with only one copy of the CH2–CH3 domain. The co-crystal structures of these monomeric Fc molecules with FcRn revealed details of the interface that could serve as a basis for building other half-life extensions.

## Results

**Disruption of CH3–CH3 interface based on structural insights.** In previous work, we generated a monomeric Fc variant, C4 (now renamed MFc1), from a rationally designed IgG4 phage library containing a set of mutations in the CH3 domain to fully stabilize the disruption of the Fc dimer interface (Fig. 1a). To extend the compatibility of these dimer-disrupting mutations with half-life extension mutations other than YTE, we initiated our effort with a test variant, T1, by replacing the YTE mutations in the CH2 domain of the monomeric Fc sequence with a new set of half-life modification mutations in CH3 around residues 432–438. This mutation set was chosen because it fit the following two criteria. First, the new mutation set could help us to explore further methods of half-life enhancement, based on the findings from a previous phage library campaign showing that it could achieve additional improvement in FcRn binding over those of the YTE mutation[25]. Second, the extensive nature of this set of mutations would be a "stress test" on the ability of the monomeric Fc to sustain disruptions in the Fc dimerization interface.

We solved the crystal structure for T1 (see Table 1 for details) and analyzed it alongside that of the dimeric IgG4 Fc and the previously solved structure of the monomeric Fc, MFc2 (or C4n, with no YTE mutations; PDB ID: 5HVW)[13]. The CH3 domains of the MFc2 and IgG4 Fc (PDB ID: 4C54) were superimposed with T1 with a root-mean-square deviation (RMSD) of ~0.7 and 0.5 Å over the Cα atoms, respectively (when excluding the last 11 residues that form an artificial chain swap in the MFc2 structure). Despite the high degree of similarity of CH3 domains, T1

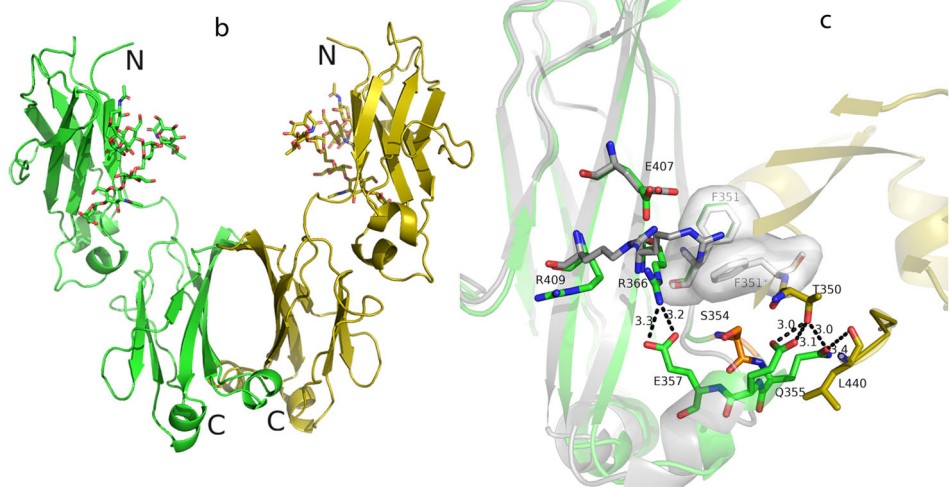

| a | | CH2 | | | CH3 | | | | | | | | | | | | |
|---|---|---|---|---|---|---|---|---|---|---|---|---|---|---|---|---|---|
| | | 252 | 254 | 256 | 351 | 354 | 366 | 395 | 405 | 407 | 432 | 433 | 434 | 436 | 437 | 438 |
| WT | | M | S | T | L | S | T | P | F | Y | L | H | N | Y | T | Q |
| Lib | | Y | T | E | X | X | X | X | Z | X | L | H | N | Y | T | Q |
| C4 | | Y | T | E | F | S | R | K | R | E | L | H | N | Y | T | Q |
| N1 | | M | S | T | F | S | R | K | R | E | C | S | W | L | C | - |
| MFC2 | | M | S | T | F | E | R | K | R | E | C | S | W | L | C | - |

**Fig. 1 Structure-guided rational design for next-generation monomeric Fc designs. a** Sequence alignment of CH2 and CH3 domains in wild-type IgG4, MFc1, and MFc2 from a previous phage library campaign, T1 variant, which replaces the YTE mutations in the MFc1 CH2 domain with a CH3 mutation set, and final sequences of MFc3 and MFc4. **b** Crystal structure of T1. **c** A closer look revealed a small set of hydrogen bonds formed in Thr350/Leu440 and Gln355/Glu356 between the crystallographic dimer chains, indicating that a possible breach of the monomeric Fc formation for T1 remained under the crystallographic conditions. The hydrogen bonds involved in the IgG4 CH3 dimer interface are now absent in the T1 crystallographic dimer due to the introduced monomer-stabilizing mutations in MFc2 (Fig. S1).

## Table 1 Crystallographic data and refinement statistics.

| | T1 | MFc2 | MFc3 | MFc4 |
|---|---|---|---|---|
| *Data collection* | | | | |
| Wavelength (Å) | 1.1279 | 1.19499 | 1.1279 | 0.97946 |
| Resolution (Å) | 35.89–2.55 (2.66–2.55)[a] | 37.74–2.30 (2.38–2.30) | 36.15–2.40 (2.49–2.40) | 36.07–2.49 (2.60–2.49) |
| Space group | C222$_1$ | C222$_1$ | C222$_1$ | C222$_1$ |
| Cell dimensions | 101.31, 101.73, 69.04 | 102.97, 110.93, 87.18 | 68.10, 122.10, 179.45 | 68.28, 122.60, 178.46 |
| *a*, *b*, *c* (Å) | | | | |
| $\alpha = \beta = \gamma = 90°$ | | | | |
| $R_{merge}$ | 0.154 (0.958) | 0.169 (1.282) | 0.149 (1.261) | 0.203 (0.847) |
| Completeness (%) | 99.9 (100.0) | 99.2 (98.2) | 99.5 (99.5) | 99.3 (94.4) |
| $I/\sigma I$ | 11.8 (2.3) | 7.4 (1.6) | 8.0 (1.6) | 4.9 (1.4) |
| Redundancy | 7.3 (7.4) | 6.8 (6.7) | 6.7 (6.7) | 6.4 (5.9) |
| CC (1/2) | 0.995 (0.800) | 0.987 (0.599) | 0.996 (0.631) | 0.984 (0.757) |
| *Refinement* | | | | |
| Unique reflections | 11,992 (1419) | 22,417 (2147) | 29,521 (3072) | 26,347 (2798) |
| $R_{work}/R_{free}$ | 0.231 (0.281) | 0.218/0.241 | 0.231/0.272 | 0.229/0.271 |
| *R.m.s. deviations* | | | | |
| Bond lengths (Å) | 0.019 | 0.007 | 0.009 | 0.006 |
| Bond angles (°) | 2.029 | 1.519 | 1.831 | 1.438 |
| Protein atoms, *n* | 1775 | 3372 | 4592 | 4579 |
| Non-protein atoms, *n* | 109 | 117 | 110 | 132 |
| *B* factor (Model/Wilson, Å$^2$) | 37.9/47.5 | 37.2/41.8 | 37.2/45.9 | 29.4/37.8 |

[a]Values in parentheses correspond to the highest resolution shell.

unexpectedly showed a formation of homodimers (Fig. 1b). The dimer interface, however, differed dramatically from that of wild-type IgG4. Visual inspection along with detailed interface analysis with PDBePISA showed that the hydrogen bonds involved in the IgG4 CH3 dimer interface were absent in the T1 dimer (Fig. S1)[30,31]. Instead, a small set of hydrogen bonds involving amino acids Thr350/Leu440 and Gln355/Glu356 were formed between the chains, indicating a possible breach of the monomeric Fc formation (Fig. 1c). Upon inspection of the newly formed T1 dimer interface superimposed with MFc2, we observed that one of the mutations in MFc2, namely, Arg366, had shifted position in its side chain. This shift allowed for the Phe351 from each chain to "reach in" and form a hydrophobic stacking interaction. In addition, the Arg366 mutation established few new hydrogen bonds that stabilized the dimer (Fig. 1c). From a close examination of the structure, we reasoned that the Arg366 shift became possible partly due to the existence of space above the Ser354 side chain. Although analytical methods showed a relatively well-behaving T1 protein (Fig. S2), the crystal structure suggests that despite the extensive disruption in dimer interface, there remained a possibility that at high protein concentrations the engineered Fc could be engaged in dimer formation.

**Structure-based engineering and characterization of binding and biophysical properties.** The newly formed interaction observed in the T1 crystal structure created an opportunity to build a more adaptable monomeric Fc molecule. From the structural inspection of the T1 dimer interface (Fig. 1c), we posited that replacing serine at position 354 with an amino acid with a bulkier side chain might prevent the Arg366 side chain from clearing the way for a hydrophobic interaction involving Phe351. This substitution could also disrupt any resulting hydrogen bond formation. Interestingly, Ser354 had been selected as one of the interfacial positions to mutate in the original phage library template, even though our first monomeric Fc construct, MFc1, had produced a mutation at every targeted position except Ser354 (Fig. 1a)[13]. We, therefore, designed a small panel of refinement mutations at position 354 to investigate whether introducing a larger side chain or electrostatic repulsion would completely stabilize the monomer formation. T1 variants containing substitutions at position 354 (T1-lib) with charged residues (R, K, D, E) and bulky polar and nonpolar residues (F, Y, P, Q, L, M) were constructed (Fig. 2a) and subsequently purified by protein A affinity chromatography. Size exclusion chromatography coupled with multi-angle light scattering (SEC-MALS) analysis revealed that nearly all the T1-lib refinement variants were monomeric (Fig. 2b–d).

To choose the most stable monomer among the T1-lib variants, we used differential scanning fluorimetry (DSF), which has been established as an orthogonal screening tool to assess thermal unfolding as a function of hydrophobic ($T_h$) residue exposure[32,33]. Thermal unfolding in the T1-lib variants was monitored and changes were observed for the transition temperature for hydrophobic exposure, $T_h$. Notably, the types of amino acid substitutions had an impact on the $T_h$ ranking, as the acidic residues (Glu and Asp) generated a higher transition temperature of up to 3 °C than basic residues (Arg and Lys) (Fig. 2e).

**Structures and properties of monomeric Fc variants with S354E (MFc3 and MFc4).** Based on SEC-MALS and DSF results, the S354E mutation was selected to be explored for the general applicability of the MFc platform. First, to confirm its compatibility with our original monomeric Fc construct, we generated MFc3, the S354E point mutant of MFc1, along with its aglycosylated variant (N297D), for crystal structural confirmation (Fig. 1a). Crystals of the aglycosylated MFc3 protein grew readily and diffracted to 2.4 Å. The solved structure demonstrated that the MFc3 molecule maintained the monomeric state at high protein concentration (Fig. 3). Superposition with the previously published monomeric Fc structure for MFc2 showed that the S354E mutation did not cause any major change in the structure of the CH2 or CH3 domains of Fc[13]. Aside from a minor change of the R405 side chain position, all the other original sets of dimer-disruptive mutations in MFc1 aligned nearly perfectly with each other. Importantly, this crystal structure demonstrated that, as designed, the glutamic acid side chain substitution at position 354 indeed protruded out and disrupted the kind of interaction we observed in T1.

In keeping with our goal of building a set of monomeric Fc variants for modulating FcRn-mediated circulation half-life, we turned our focus to functional and structural characterization of the FcRn interaction with our monomeric Fc molecules. Using recombinant FcRn binding analysis, we found that MFc3 did not differ significantly from MFc1, showing an equilibrium dissociation constant ($K_D$) of approximately 300 nM (Table 2). This finding suggested that the S354E substitution did not alter the interaction with FcRn. To confirm this mode of binding, the MFc3/FcRn complex was prepared at low pH and was subsequently purified and crystallized, and diffraction data were collected to a resolution of 2.6 Å (Fig. 4a). The closest Fc–FcRn complex structure available for comparison was the one solved by our group, which consists of human IgG1 Fc (with the YTE set of mutations) bound to human FcRn in complex with human serum albumin at 3.8 Å (PDB ID: 4N0U)[29]. Comparison of the two interfaces showed that the mode of interaction remained nearly identical, having minor differences that most likely arose from the difference in resolving the side chains in the electron density maps. This structural and functional invariance of the Fc–FcRn interaction was expected because the S354E mutation is more than 20 Å away from the FcRn interface.

To evaluate whether the S354E mutation could indeed be used to bring T1 to a monomeric state, we crystallized T1-S354E (MFc4) in complex with FcRn (Fig. 4b). The resolved MFc4/FcRn complex structure demonstrated that MFc4 was indeed monomeric. Despite the fact that the crystallization condition search for the MFc3/FcRn and MFc4/FcRn complexes was done independently, the crystals grew from the same condition and demonstrated nearly identical cell parameters and space group. As expected from previous work, the sequence changes from MFc3 to MFc4 resulted in a dramatic increase in recombinant FcRn binding affinity, from 300 to 5 nM (Table 2)[25]. The overall structures of the MFc4/FcRn and MFc3/FcRn complexes superimpose with an RMSD of 0.35 Å over 3700 non-hydrogen atoms, suggesting a high degree of similarity. A comparative view of the MFc3/FcRn and MFc4/FcRn complex structures revealed distinct details of the Fc–FcRn interaction and provided structural explanations for the differences we observed in FcRn binding. The newly introduced Tyr434 and Leu436 in MFc4 increased the interface area from 500 Å² to nearly 600 Å², while adding some hydrophobicity to the interaction (Fig. 4c, d). Residue Leu135 in FcRn, which had been known to be a contributor of hydrophobicity at the Fc/FcRn interface[29], was now more involved in the presence of the Tyr434 and Leu436 in MFc4.

Using PDBePISA analysis, we captured the energy contributions from individual residues in the binding pockets of MFc3/FcRn versus MFc4/FcRn with a differential solvation energy heat map (Fig. 4e). As a measurement of the extent of the binding interface, the differential solvation energy calculations reflect the part of the structure's surface that becomes inaccessible to solvent[30,31]. The strongest contributors to the binding interface

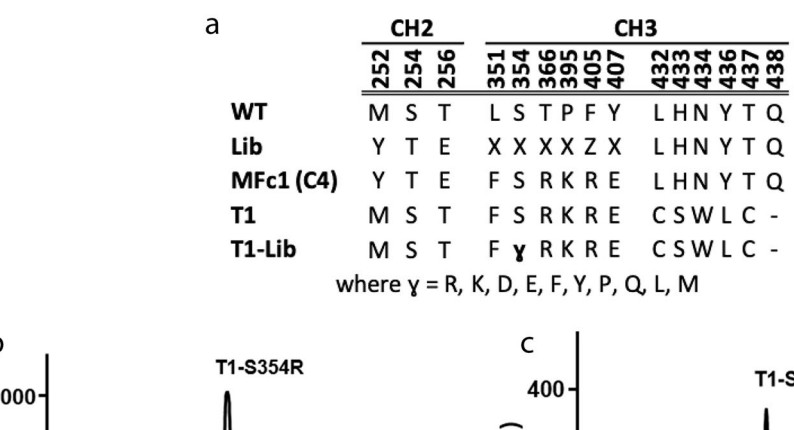

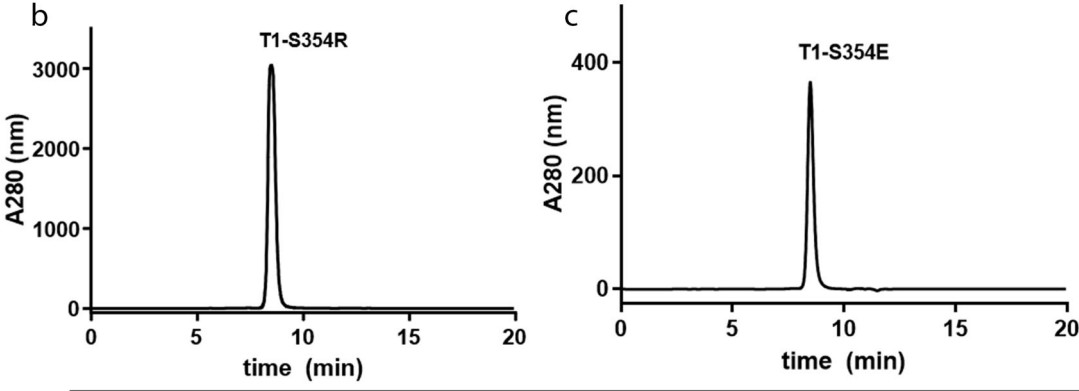

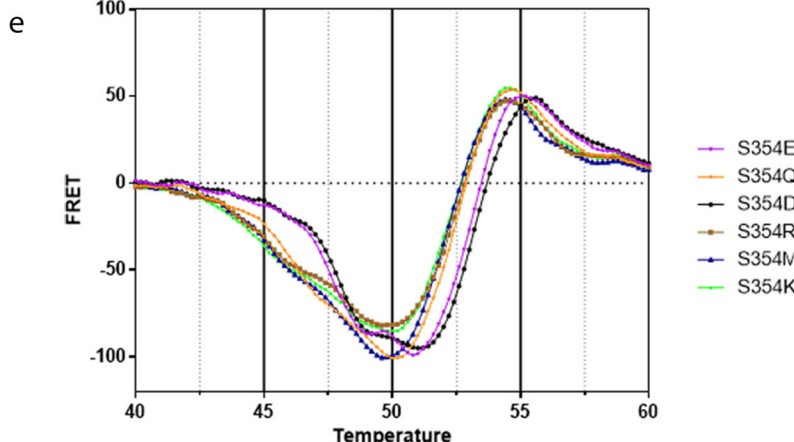

**Fig. 2 Rational engineering to enable next-generation MFc designs. a** Sequence alignment of CH2 and CH3 domains in wild-type IgG4, T1, and T1-lib, a panel of point-mutation variants targeting residue S354. **b**–**d** Representative SEC-MALS analysis results showing the T1 point mutants demonstrating a molecular weight of approximately 26–27 kDa with good homogeneity; **e** DSF comparisons among the T1 refinement mutants to identify S354E and S354D with moderately higher thermal stability.

from MFc4 were found to be Ile253, Tyr434, and Leu436, in contrast to residues Ile253 and Thr254 in MFc3. The total solvation energy change was 5.09 kcal/mol for MFc4, a notably stronger hydrophobic interaction than MFc3, which had a total $\Delta iG$ of 3.14 kcal/mol (Figs. 4e and S3)[31]. As a comparison, we ran the same calculation on the only other available human YTE IgG1 Fc–FcRn complex and found a similar $\Delta iG$ of

2.67 kcal/mol, which was consistent with the binding affinity measurements.

The MFc4/FcRn complex structure also provided us with structural insight into the wild-type human Fc/FcRn interaction, which had never been available before for two reasons. First, wild-type human Fc/FcRn interaction is relatively weak, making complex purification difficult, if not impossible. Second, the

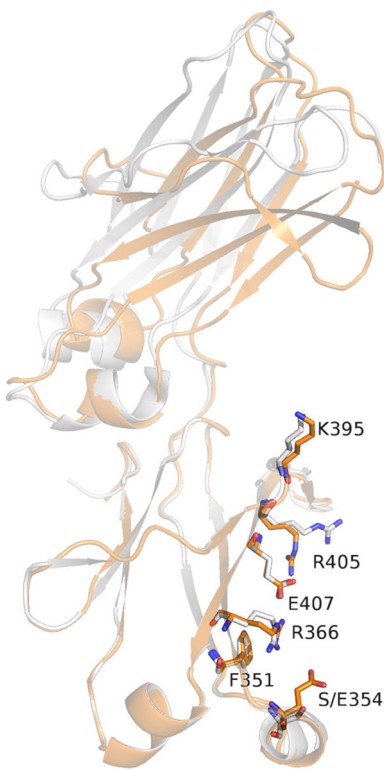

**Fig. 3 Crystal structure of a new monomeric Fc, MFc3.** MFc3 (Fig. 1a) in the aglycosylated form (N297D) was crystallized in the C222$_1$ space group with cell parameters $a = 102.97$ Å, $b = 110.93$ Å, $c = 87.18$ Å, Matthews parameter $V_M = 2.49$ Å$^3$/Da with two polypeptides in the asymmetric part of the unit cell, and solvent content of 50.62%. The diffraction limit of the crystal was 2.3 Å. The structure was solved by a molecular replacement method implemented in MolRep within the CCP4 suite of programs, using a model of C4n (PDB ID: 5HVW) as a starting model[33,35]. Refinement was performed using Refmac5 while manual building/rebuilding was performed using "O."[39,42] Data collection and final refinement statistics are shown in Table 1. Superposition of MFc3 (orange) with a previously resolved structure of MFc2 (or C4n) (light purple) shows that both maintained a similar monomeric Fc structure, and the S354E mutation did not cause any notable change in the Fc domain structures. As designed, the glutamate side-chain due to the S354E mutation protruded into any possible dimer interaction observed in T1 (Fig. 1c).

propensity of dimeric Fc for crystallization (hence the name "fragment crystallizable") is so high that most attempts yield crystals containing just Fc. Prior to the availability of our stable monomeric Fc variants, we had to rely on a rat Fc or FcYTE for the improved affinity between Fc and FcRn, along with albumin to disrupt the Fc crystal lattice formation, to increase the residence time of the bound state for crystal formation[29]. Now, with the help of MFc3 and MFc4 structural complexes with FcRn, with their two distal sets of mutations, we were able to gain a better understanding between wild-type binding interfaces and their mutation sets. For example, at the wild-type interface around residues 252, 254, and 256, where the YTE (M252Y/S254T/T256E) mutations reside, the interface solvation energy map showed that Met252 had an only moderate contribution to the FcRn interface, whereas Thr256 was not actually involved (Fig. 4e). The most significant FcRn interaction from the 252–254–256 loop was provided by Ser254. This analysis also explains why the YTE set of mutations could improve affinity between Fc and FcRn and why the strongest contribution came from its S254T substitution.

**Table 2 Equilibrium binding of monomeric Fc variants to human FcRn in a recombinant 1:1 binding format.**

| Construct | $K_D$ (nM) at pH 6 |
| --- | --- |
| MFc1 (C4) | 370 |
| MFc2 (C4n) | 3560 |
| MFc3 | 330 |
| MFc4 | 5 |
| IgG1 Fc | 300 |

**Construction of a monomeric bispecific molecule.** These monomeric Fc molecules can be easily used as building blocks for designing monovalent, dual-targeting Fc fusion proteins. Previously we used the MFc1 variant to generate an onartuzumab Fab–MFc1 fusion protein[8,13]. In the present study, we designed the first example of monovalent bispecific targeting molecules with the monomeric Fc constructs. With a single-plasmid construction, we attached the same Fab domain from onartuzumab to the N-terminus of MFc1 or MFc4, along with a C-terminal single-chain variable fragment (scFv) of an antibody targeting programmed cell death ligand 1 (PD-L1) (Fig. 5a). The constructs were transfected for transient expression in HEK293 suspension cultures (expression titers around 90 mg/L), and the protein was subsequently purified in a single-step protein A purification. SEC-MALS analysis suggested that the protein was monodisperse and had the expected molecular weight of 100 kDa (Fig. 5b). The dual-targeting activity of the monomeric bispecific molecules was confirmed on an Octet platform in a sandwich format (Fig. 5c).

**Improved in vivo half-life with next-generation monomeric Fc.** We achieved significant improvement of in vitro FcRn binding in the MFc4 variant as compared with MFc1. We also wanted to evaluate whether this improvement would translate into the enhancement of in vivo serum half-life. The newly generated Fab-MFc4-scFv and Fab-MFc1-scFv, with molecular mass (100 kDa) well above the typical renal filtration clearance size (~60 kDa)[34], are ideal molecules to evaluate the implications of improving the in vivo half-life of FcRn binding. We carried out in vivo pharmacokinetic (PK) studies in hemizygous human FcRn (TG276) transgenic mice. This mouse model is a well-studied model that reflects demonstrable PK impact on human FcRn binding from Fc mutations and a standard IgG1 with a serum half-life of approximately 18 h[13,25,35]. Mice were dosed with fusion proteins at 2.5 mg/kg, and serum protein concentrations were determined by enzyme-linked immunosorbent assay (ELISA). The Fab-MFc4-scFv bispecific protein had higher serum levels than Fab-MFc1-scFv (Fig. 6a). PK parameters were analyzed and determined, showing that the clearance rate and terminal half-life for Fab-MFc4-scFv were markedly greater than those of Fab-MFc1-scFv, by nearly twofold. This indicates that the stronger MFc4-mediated human FcRn binding contributes to enhanced serum protein recycling compared with MFc1. As expected, the increased molecular size contributed to reduced clearance from Fab-MFc1 to the bispecific molecule Fab-MFc1-scFv.

## Discussion
Monovalent antibodies or fusion proteins based on monomeric Fc have the potential to confer IgG-like serum properties to an expanded class of protein therapeutics. Following the successful engineering of a stable monomeric Fc, we took on the challenge to build and expand the utility of the MFc platform to achieve several key properties.

First, we wanted to establish a more universal monomeric Fc molecule that would accommodate alternative Fc mutations for

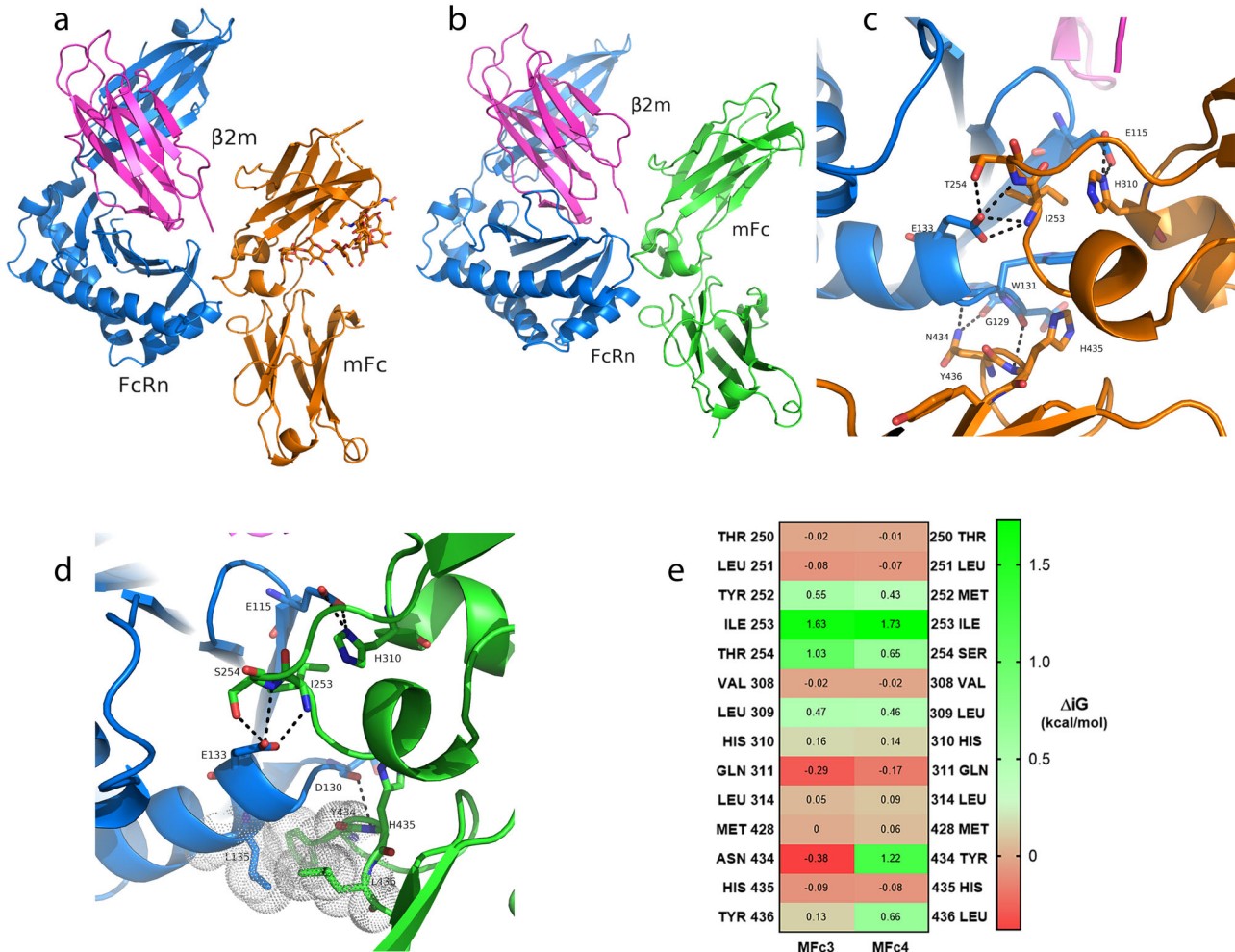

**Fig. 4 Structural interrogation of FcRn interaction with MFc3 and MFc4. a** MFc3 (orange) and FcRn (blue)/β2-macroglobulin (pink) complex were crystallized in the C2221 space group with cell parameters $a = 68.10$ Å, $b = 122.10$ Å, $c = 179.45$ Å, Matthews parameter $V_M = 2.87$ Å$^3$/Da, solvent content of 57.2%, and one complex per asymmetric part of the unit cell. **b** Binding interface between MFc3 (orange) and FcRn (blue). Hydrogen bonds are indicated by dashed lines. **c** MFc4 (green) and FcRn (blue)/β2-macroglobulin (pink) complex were crystallized. **d** Binding interface between MFc4 (green) and FcRn (blue). Hydrogen bonds are indicated by dashed lines. **e** Heat map showing the differential solvation energy $\Delta iG$ (kcal/mol) contribution from each MFc residue involved in the receptor-binding interface. Higher positive values indicate a stronger solvation effect from the monomeric Fc bound surface.

half-life tuning, including the potential for further half-life extension. Second, we sought to demonstrate that the MFc platform could indeed sustain the monomeric state and be stable for bispecific molecular targeting, the desired targeting strategy to enable novel therapeutic applications including immuno-oncology and receptor-mediated transcytosis[4–7,9]. A monovalent bispecific drug format built around a monomeric Fc could offer the advantages of complete ablation of effector function and reduction of toxicity and off-target sink, a design feature that is optimal for T-cell and other immune-cell engagers. Finally, we aimed to validate a computational approach to enable the development of future monomeric Fc designs. To accomplish these goals, we attempted to harness the power of structure-guided molecular design.

We had previously shown that the YTE mutation in MFc1 was able to improve FcRn binding affinity to offset the reduction in binding avidity[13]. Based on the output from a prior Fc phage library and engineering work, we identified a mutation set, T1 (Fig. 1), with promising enhancement in FcRn binding at pH 6.0. This offered us an ideal test case to evaluate the adaptability of the monomer-stabilizing mutations to half-life mutations in the CH3 domain, as opposed to the YTE mutations from the CH2 domain.

The crystal structure of the T1 protein suggested that the engineered Fc could be engaged in a newly packed dimer formation at high protein concentrations. Although we did not find any direct contribution from the new set of half-life extension mutations on this new form of Fc dimerization, we believe that those mutations play a role in allowing for an induced dimerization under tight packing conditions. A close examination of this dimer interface guided our rational design to enlarge the side chain of residue at position 354. From the set of mutations to replace the serine residue, we chose glutamic acid based on its superior thermal stability profile (Fig. 2). Using solution analytical methods and crystallography, we demonstrated that S354E was able to sustain a monomeric Fc structure, and the glutamic acid side chain protruded out as designed (Fig. 3). The confirmation of the adaptability and activity of S354E for monomeric Fc structures was observed in co-crystal structures when MFc3 and MFc4 maintained the monomeric state and retained extensive engagement with FcRn (Fig. 4a, c). This is to our knowledge the first time it has been demonstrated that co-crystal structures can be easily generated to interrogate the Fc-FcRn interface and that quantifiable differences exist in those binding interactions. Using a differential solvation energy calculation for the binding

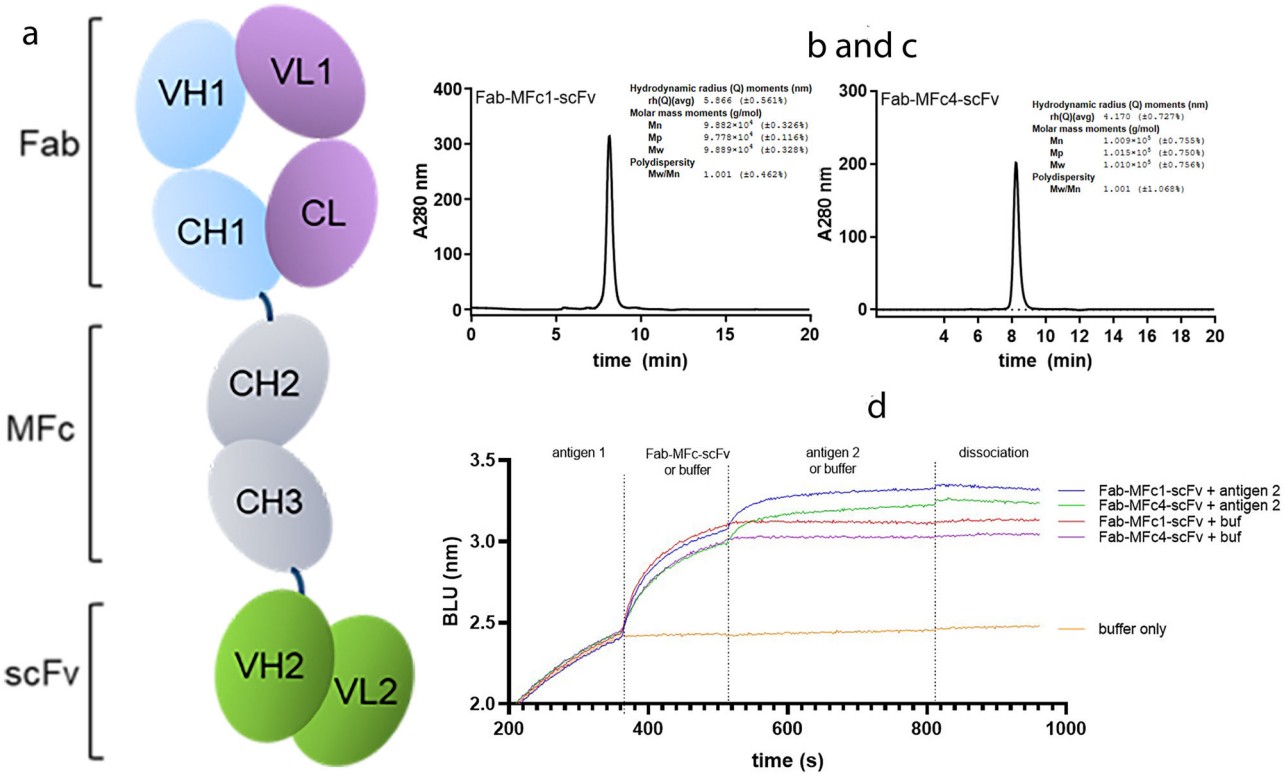

**Fig. 5 Generation of monomeric Fc bispecific antibodies. a** Cartoon of a monomeric Fc-based monovalent bispecific antibody. **b**, **c** SEC-MALS analysis of Fab-MFc1-scFv and Fab-MFc4-scFv showing the measured molecular weight of approximately 100 kDa, with a polydispersity of 1.001. **d** Concurrent binding analysis using biolayer interferometry demonstrated expected binding activity from both the Fab and scFv moieties to recombinant antigens.

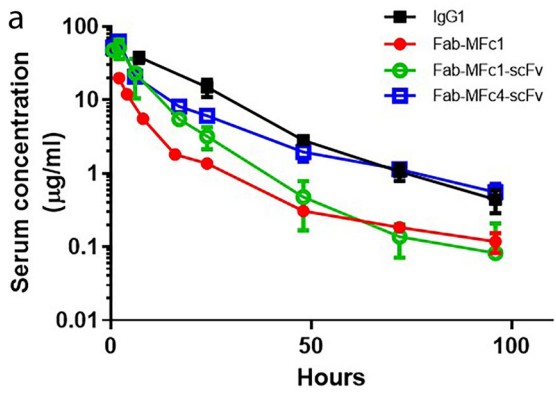

**b**

| Construct | Dose (mg/kg) | $C_{max}$ (μg/mL) | $AUC_{INF}$ (hr*μg/mL) | CL (mL/hr/kg) | $T_{1/2}$ (hr) |
|---|---|---|---|---|---|
| Fab-MFc1-scFv | 5 | 51 | 436 | 11 | 12.6 |
| Fab-MFc4-scFv | 5 | 62 | 590 | 8 | 26.4 |
| Fab-MFc1 | 2.5 | 20 | 160 | 16 | 21.0 |
| IgG1 control | 2.5 | 38 | 791 | 3 | 17.9 |

**Fig. 6 In vivo mouse PK analysis of monomeric Fc–bispecific antibodies. a** hFcRn transgenic mouse serum clearance curves are plotted for Fab-MFc1-scFv, Fab-MFc4-scFv, Fab-MFc1, and IgG1 control ($n = 4$ animals per time point), based on concurrent Fab and Fc domain binding. **b** PK parameters were determined by noncompartmental analysis with model 201. $AUC_{INF}$ = area under the concentration-time curve for the plasma concentration versus time graph from time 0 to infinity; CL clearance, $C_{max}$ peak concentration, $t_{1/2}$ = terminal half-life.

interface, we were able to quantify the enhanced FcRn binding due to extended hydrophobic interaction in MFc4 (Fig. 4b–e).

The availability of monomeric Fc molecules with variable FcRn binding capabilities allowed us to validate the use of the MFc platform for building monovalent bispecific molecular targeting. We generated Fab-MFc-scFv molecules with both MFc1 and MFc4 that had demonstrable monomer conformational purity and dual-targeting activity. With a molecular size well above the renal filtration cutoff, these molecules are well-suited for evaluating the in vivo pharmacokinetic consequences of improving FcRn binding. We found that the Fab-MFc4-scFv bispecific molecule sustained a higher serum level than Fab-MFc1-scFv, nearing that of a standard IgG1 antibody. These results also indicate that these MFc constructs can contribute to tunable serum protein recycling and can provide a versatile bispecific platform for the development of next-wave technology to support therapeutic advances.

The exposure of the CH3 domain due to the disruption of the Fc dimer is not a new phenomenon. IgG4 backbone had been chosen for our MFc platform due to the ready engagement of Fab-arm exchange of the IgG4 molecules which alternates the Fc between a dimer–monomer equilibrium[36]. In addition, to launch a strong and specific immune response, T-cell epitopes need to be processed and presented by antigen-presenting cells[37]. To mitigate the concern that the newly introduced mutations may form a novel T cell epitope, an in silico T cell epitope prediction around the monomer-forming mutations in MFc1 and MFc4 (Fig. S4) was performed. We observed an overall low predicted binding rate for these mutations (Fig. S4C), indicating a lower immunogenicity risk[38]. Nevertheless, given the promise of these MFc-based molecules for therapeutic development, we hope to validate their PK properties in non-human primates, along with other developability assessments including immunogenicity.

As the field of drug discovery witnesses an accelerated pace of development, from chemotherapy to targeted therapy, to cancer immunotherapy, many innovations have been further driving immunotherapeutics from immune checkpoint inhibitors toward synthetic immunity, such as chimeric antigen receptor T-cell therapy and immune-cell engagers[39]. A complete silencing of effector function could offer reduced toxicity and an off-target sink, a design feature that could be desired for T-cell and other immune-cell engagers. Previous work demonstrated the bivalent requirement for FcγRs engagement[40], and that a monomeric Fc did not bind to FcγRs[13]. We have further confirmed that the OnartFab-MFc molecule did not trigger ADCC using both low- and high-expressing target cells, while its dimeric Fc counterpart did (Fig. S5). In addition to immuno-oncology applications, it had also been previously reported that monovalent transferrin receptor-mediated transcytosis conferred a different cellular trafficking route with reduced lysosomal degradation[9]. The increased serum half-life for MFc4 suggests a more efficient cellular transcytosis and recycling via FcRn could be involved. Consistent with this, human endothelial cell-based recycling assays had previously demonstrated that longer half-lives in human FcRn transgenic mice correlated with enhanced rescue from intracellular degradation[41]. Assessing the pharmacokinetics of MFc4 in non-human primates will be a desirable next step to further validate the enhanced recycling. Our work has also enabled future studies to further explore the relationship between affinity, cellular trafficking, and serum half-life for molecules with monovalent FcRn binding. The ultimate success of curative therapeutics will rely on many collaborative efforts to address the balances of potency versus toxicity and serum half-life versus tissue penetration, in conjunction with a deepened understanding of immunity and translational sciences. With the ability and flexibility to present monovalent bispecific targeting motifs to sidestep any undesired Fc receptor-mediated cytotoxicity and off-target sinks while mimicking IgG-like serum properties, the MFc platform presents a timely possibility to further expand the exploration of immune-cell engagers.

## Methods

**Ethics statement.** The protocol (MI-13-0012) requiring the use of animals in these studies was reviewed and approved by AstraZeneca's Institutional Animal Care and Use Committee and complies with the animal welfare standards of the U.S. Department of Agriculture, the *Guide for the Care and Use of Laboratory Animals*, and the Association for Assessment and Accreditation of Laboratory Animal Care.

**Antibody cloning, expression, and purification.** All antibody positions are listed according to the Kabat numbering convention for the variable domains and EU numbering convention for the CH2–CH3 domain[42,43]. All chemicals were of analytical grade. Oligonucleotides were purchased from Eurofins MWG Operon (Louisville, KY). Constructs were generated with the In-Fusion HD cloning kit from Takara Bio (Mountain View, CA), encoding variable heavy chain and variable light chain sequences into an in-house IgG1 mammalian expression vector. Point mutations were introduced by site-directed mutagenesis, using the QuikChange Multi Lightning mutagenesis kit (Agilent Technologies, Santa Clara, CA).

Variant constructs were transfected transiently in human embryonic kidney cell line HEK293FT, using 293Fectin transfection reagent (Life Technologies, Carlsbad, CA). Cells were grown in FreeStyle 293-F Expression Medium (Life Technologies). The expressed proteins were purified from supernatants by affinity chromatography on a HiTrap Protein A column (GE Healthcare Life Sciences, Marlborough, MA). Proteins were eluted with Pierce IgG Elution Buffer (Thermo Fisher Scientific, Waltham, MA), neutralized with 1 M Tris at pH 8.0, then dialyzed overnight into phosphate-buffered saline (PBS), pH 7.2.

**SEC-MALS and analytical ultracentrifugation.** Purified Fc clones and fusion proteins at concentrations of 1 mg/mL or greater were analyzed by SEC-MALS, as previously reported[13]. The samples were run on an 1100 HPLC instrument (Agilent, Santa Clara, CA), using a TSK-GEL G2000SWXL column (Tosoh Biosciences, Tokyo, Japan), and eluted isocratically in PBS at a flow rate of 1 mL/min for 20 min. Eluted proteins were detected with ultraviolet absorbance at a wavelength of 280 nm, and analyzed with molecular-weight standards ranging from 10 to 500 kDa (Bio-Rad, Hercules, CA). In-line SEC-MALS was performed on a Dawn

Heleos II MALS with an Optilab Rex refractometer (Wyatt Technologies, Santa Barbara, CA). The molecular mass of each protein within a defined chromatographic peak was calculated on the Astra software (Version 6.1, Wyatt Technologies).

Analytical ultracentrifugation analysis was run as in the previous studies[13]. Samples and reference buffer were loaded into 12-mm double-sector cells with Epon centerpieces on an Optima XL-I centrifuge set to 20 °C (Beckman-Coulter, Indianapolis, IN). An-50 Ti rotor was used for ultracentrifugation at 50,000 rpm. The sedimentation data collected at 280 nm from scans 2 to 160 were analyzed with the Sedfit software (version 16.1c) to generate c(s) distributions[44,45]. The partial specific volume was set to 0.73 mL/g. Solution density and viscosity values for PBS were set to 1.00523 g/mL and 1.019 mPa s, respectively, using the calculated value from the Sednterp program (version 20130813)[46]. Based on the Svedberg equation, a monomeric Fc with a molecular mass of 27 kDa is expected to have a sedimentation coefficient of 1.7–2.4 S (Svedberg units), assuming a frictional ratio of 1.3–1.8 (globular to extended shape).

**Crystallization, data collection, and structure determination.** Prior to crystallization, protein A purified T1, MFc3, and MFc4 were further purified by ion-exchange chromatography on a Q HP 5 mL prepacked column (GE Healthcare Life Sciences) equilibrated with 25 mM Tris-HCl buffer at pH 8 further purified by SEC, using a Superdex 200 Increase 10/300 GL column (GE Healthcare Life Sciences) pre-equilibrated with 25 mM Tris-HCl, pH 8, and 100 mM NaCl. The cultured media of recombinant heterodimeric FcRn after harvest was pH adjusted for affinity purification on an IgG Sepharose column (GE Healthcare Life Sciences). After FcRn was purified on a Q HP column (GE Healthcare Life Sciences), it was dialyzed into 30 mM sodium acetate buffer at pH 5.2 and complexed with MFc3 and MFc4 at a 1% molar deficit of FcRn, and the complex was purified by SEC using the same Superdex 200 column equilibrated with 30 mM sodium acetate, pH 5.2, and 100 mM NaCl. The complex composition was confirmed by sodium dodecyl sulfate-polyacrylamide gel electrophoresis.

Initial crystallization trials for all proteins and protein complexes were carried out by the sitting-drop vapor-diffusion method at 20 °C. The crystallization drops were dispensed in 96-well crystallization plates (Intelli-plate 102-0001-20; Art Robbins Instruments, Sunnyvale, CA) by a Phoenix robot (Art Robbins Instruments) and were composed of equal volumes of protein and reservoir buffer. For crystallization of T1 and MFc3 by themselves, we used commercially available screens (Hampton Research, Aliso Viejo, CA; Molecular Dimensions, Suffolk, UK). For the crystallization of the FcRn-complexed proteins, we generated a new screen consisting of a combination of the low-pH conditions contained in commercially available screens. Diffraction quality crystals were grown in the crystallization optimization step in hanging drop format from the following crystallization solutions: T1: 0.01 M zinc sulfate heptahydrate; 0.1 M morpholineethanesulfonic acid (MES) monohydrate, pH 6.5, and 25% (w/v) PEG 550 MME at a protein concentration of 5.5 mg/mL. MFc4/FcRn complex: 0.2 M magnesium chloride hexahydrate, 1 M sodium iodide, 0.1 M MES, pH 6 and 20% PEG 6000 at a protein concentration of 6.35 mg/mL. MFc3/FcRn complex: 0.2 M magnesium chloride hexahydrate, 30% 1,5-diaminopentane dihydrochloride, 0.1 M MES, pH 6 and 20% PEG 6000 at a protein concentration of 6 mg/mL. The crystals for MFc3 were harvested directly from the original sitting drop plates from a condition consisting of 0.8% anesthetic alkaloids (2% w/v lidocaine hydrochloride monohydrate, 2% w/v procaine hydrochloride, 2% w/v proparacaine hydrochloride, 2% w/v tetracaine hydrochloride), 0.1 M MOPS (acid) and sodium HEPES pH 7.5, and a 50% v/v mix of precipitants (40% v/v ethylene glycol, 20% v/v PEG 8000) at a protein concentration of 7 mg/mL. All crystals harvested for X-ray analysis were flash-cooled by dipping in liquid nitrogen. Diffraction data were collected from single crystals on beamline BL9-2 of a Stanford Synchrotron Radiation Lightsource equipped with a Pilatus 6 M PAD detector (Paul Scherer Institute, Villigen, Switzerland) over an oscillation range of 180°, an increment of 0.5°, and a 0.8-s exposure per image. Diffraction data were processed with the XDS program[47]. All crystallographic calculations were carried out with the CCP4 software suite (version 7.0)[48]. The molecular replacement procedure was performed by using the Molrep program[49]. Structure refinement was performed with Refmac5, and model adjustments were carried out with the "O" program[50,51]. Figures with structures were generated with PyMOL (Schrödinger, New York, NY).

**Octet binding analysis.** Binding measurements of the monomeric Fc and its fusion proteins to in-house purified recombinant human FcRn were carried out on an Octet384 instrument (Sartorius Lab Instruments, Goettingen, Germany), as previously described[13]. Biotinylated FcRn were captured on streptavidin biosensors at 1 μg/mL in PBS buffer (pH 7.4) or 100 mM MES buffer (pH 6.0), with 3 mg/mL bovine serum albumin, 0.05% Tween 20. Association of the different Fc variants or Fc fusion constructs, in 1:3-fold serial dilution, to the loaded biosensors were measured, followed by dissociation measurements. Octet software (version 7.2) was used to calculate steady-state apparent affinities ($K_D$) from a nonlinear fit based on the 1:1 binding kinetic model of the data. Concurrent binding measurements of Fab-MFc-scFv molecules to recombinant antigen proteins were also performed. Biotinylated cMet protein was captured at 5 μg/mL on streptavidin biosensors in PBS buffer, pH 7.2, with 1× kinetics buffer. The binding steps included 300 nM

Fab-MFc-scFv with buffer control, followed by binding to the second antigen along with buffer control.

**In vivo PK in hFcRn transgenic mice**. Human FcRn transgenic mice used in this study are the F1 cross of murine FcRn-deficient B6.129×1-*Fcgrttm1Dcr/DcrJ* and human FcRn cDNA transgenic line B6.Cg-Fcgrttm1Dcr Tg (CAG-FCGRT) 276 *Dcr/DcrJ*, using a previously established protocol[13]. Sex-matched (6–16-week-old) mice were dosed intravenously with 2.5 mg/kg monomeric Fc fusion proteins on day 0. Eight mice were used per protein, and two groups of mice (groups A and B) were bled at alternate time points. A quantitative ELISA was used to monitor the serum concentrations at different time points using 96-well plates coated with 2 μg/mL cMet extracellular domain. Goat anti-human Fc-specific horseradish peroxidase-conjugated antibody at 1:10$^4$ dilution (Jackson ImmunoResearch Laboratories, West Grove, PA) was used for detection. The linear portions of standard curves were used to quantify human anti-cMet fusion proteins in the serum samples. Non-compartmental PK data analysis was performed with Phoenix 64 WinNonlin 6.3 (Pharsight, Mountain View, CA). PK parameters were summarized statistically and presented as mean. The maximum observed peak plasma concentration ($C_{max}$) was determined on the observed data using WinNonlin. The terminal elimination half-life ($T_{1/2}$) was calculated as $\ln(2)/\lambda z$, where $\lambda z$ is the slope of the terminal portion of the natural-log concentration-time curve, determined by linear regression of at least the last three time points. The systemic exposure was determined by calculating the area under the curve (AUC) for the plasma concentration versus time graph from the start of dosing to the time of last measurable concentration, using the linear/log trapezoidal rule. AUC for the plasma concentration versus time graph from time 0 to infinity ($AUC_{INF}$) was calculated as $AUC_{last} + C_{last/\lambda z}$, where $C_{last}$ is the last quantifiable concentration. Clearance (CL) was calculated by dose/$AUC_{INF}$.

**Reporting summary**. Further information on research design is available in the Nature Research Reporting Summary linked to this article.

## Data and materials availability

All data needed to evaluate the conclusions in the paper are present in the paper, the Supplementary Information file as well as Supplementary Data 1 spreadsheet. Additional data related to this paper may be requested from the authors. Coordinates and structure factors are deposited in the Protein Data Bank under IDs 6WIB, 6WMH, 6WNA, and 6WOL.

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

## Acknowledgements

We thank Dr. Qing Li for assisting with the in vivo PK study design and Deborah J. Shuman for the critical reading of the manuscript. Use of the Stanford Synchrotron Radiation Lightsource, SLAC National Accelerator Laboratory, is supported by the U.S. Department of Energy, Office of Science, Office of Basic Energy Sciences under Contract No. DE-AC02-76SF00515. The SSRL Structural Molecular Biology Program is supported by the DOE Office of Biological and Environmental Research, and by the National Institutes of Health, National Institute of General Medical Sciences (including P41GM103393). Synchrotron X-ray diffraction data collection and processing services were provided by Accelero Biostructures, Inc., San Francisco, CA.

## Author contributions

L.S. designed, optimized, and characterized mutants and conducted experimental guidance and data analysis; N.D. performed crystallization of monomeric Fc variants and FcRn; N.H. performed protein purification and in vivo, PK ELISA analysis; K.C. performed protein purification and characterization; K.R. conducted SEC-MALS analysis; V.O. designed mutants solved the crystal structures and performed structure refinement and analysis; S.D. conducted AUC experiments; Y.J. performed in vivo PK modeling analysis; M.J.B. generated the T1 mutation and reviewed the data; M.D., H.W., and W.F.D. supervised the project and reviewed data; L.S. and V.O. conceived and supervised the project, designed experiments, reviewed data, and wrote the paper. All authors approved the final version of the paper.

## Competing interests

The study was funded by AstraZeneca. All authors were employees of AstraZeneca at the time the research was conducted, with stock ownership and/or stock interests or options in the company.
