## [Peer Review File · Communications Biology]

Reviewers' comments:

Reviewer #1 (Remarks to the Author):

Thorough and well written paper describing new monomeric Fc variants with improved FcRn binding with extensive biochemical and structural analyses and further presenting a monovalent Fc-based bispecific format (Fab-MFc4-scfv). Experiments are carefully performed, the data analysis and conclusions are sound and the constructs engineered and characterized of significant potential utility for biotechnology applications. Publication is recommended with minor changes:

- 1) The transition between the T1 variants and mFc3 and mFc4 is a bit confusing. Clarify what exactly are these two antibodies.
- 2) Do mFc3/mFc4 show binding to FcRn at physiological pH in a manner similar to YTE (on which they are partly based)? I think that would be important to show.
- 3) Report Vss?
- 4) Finally can the authors comment on expression yields and also on stability to storage (stress test assays perhaps) as these features are of interest to readers interested in therapeutic applications of this technology?

Reviewer #2 (Remarks to the Author):

This manuscript describes novel mutations of monomeric Fc constructs intended to facilitate the crystallization of Fc-FcRn complexes and to be therapeutically applied. They present convincing data for the mutations identified that can be used in stable monomeric Fc constructs with tunable PK properties. These informations and this approach is useful for the Fc engineering community. The presentation of the work is solid, described in sufficient detail and I have no major comments or concerns

Reviewer #3 (Remarks to the Author):

Review Manuscript#: COMMSBIO-20-1189-T

Title: In vivo pharmacokinetic enhancement of monomeric Fc and monovalent bispecific designs through structural guidance

Major claims of the paper:

The authors of the Manuscript (COMMSBIO-20-1189-T) describe in a scientific interesting way the antibody scaffold engineering, necessary to design an FcRn binding active monomeric Fc construct. The description of their structural guided approach to reach improvement in in vivo serum half-life is straightforward.

Starting from a previously known monomeric Fc variant, the work presented, describe additional analysis of the Fc dimer interface and the interplay of the FcRn binding CH3 area.

The authors have tested the influence of the residue S354 on thermostability, solved the crystal structure of two variants combining the new identified favorable S354E mutation with the previous known YTE mutation(MFc3) and a new N434Y & Y436L combination. (MFc4).

These variants have been tested by SEC-MALS & analytical ultracentrifugation, FcRn binding analysis utilizing Octet and in vivo PK in hFcRn transgenic mice.

This is a readable follow up of a previous research article from 2016, authored by the same group.

To publish in a Journal, addressing a broad audience, I am missing data describing the new behavior/biology behind a monomeric Fc such as the expected impact to the human immune system

1)What is the difference in cellular trafficking of such constructs? Is the routing of a monomeric Fc, possessing improved FcRn binding affinities, similar to a weaker binding symmetric WT IgG1 or can it be found in other cellular compartments?

2)Is the pH dependent cellular recycling mechanism affected or not? Testing in a cellular FcRn assay would provide more insight on this aspect, ideally different FcRn expression levels and in presence of relevant human IgG levels. Determining recycling and transcytosis efficiency.

3)Serum IgG competition. How does such a molecule behave in a more human like situation? (e.g. in the mouse model utilized in Lee CH, et al. Nat Commun. 2019. PMID: 31695028 or a IgG competition model (Stapleton NM, Nat Commun. 2011 Dec 20;2:599. doi: 10.1038/ncomms1608))

4)How does the Fab and scFv of this monomeric Fc affect FcRn binding in contrast to a one armed dimeric Fc plus scFv construct?

5)Is there an immunogenic impact expected of an engineered CH3 domain that is usually not accessible?

6)A monomeric Fc is usually not expected to bind Fc gamma receptors, but is this true for high target expressing cells? An experiment that proofs if a weak Fc gamma receptor binding is restored by high avidity on a highly Fab-mFc decorated target cell will be helpful to answer such a question. Especially of interest because the monomeric Fc is stabilized.

I would not recommend the publication of this manuscript in its current form; however, I would reconsider reviewing the manuscript after substantial addition of data addressing my questions above.

Kind regards Tilman Schlothauer

In vivo pharmacokinetic enhancement of monomeric Fc and monovalent bispecific designs through structural guidance

Referee #1: Engineering of antibody therapeutics

Referee #2: Antibody therapy

Referee #3: Antibody therapy

Reviewers' comments:

Reviewer #1 (Remarks to the Author):

Thorough and well written paper describing new monomeric Fc variants with improved FcRn binding with extensive biochemical and structural analyses and further presenting a monovalent Fc-based bispecific format (Fab-MFc4-scfv). Experiments are carefully performed, the data analysis and conclusions are sound and the constructs engineered and characterized of significant potential utility for biotechnology applications. Publication is recommended with minor changes:

1) The transition between the T1 variants and mFc3 and mFc4 is a bit confusing. Clarify what exactly are these two antibodies.

We appreciate the reviewer's request for explaining these variants. We have added more clarification to the description in two places. On Page 6 (Lines 7-8) we have described the purpose of the T1 variant as a test case toward the next-generation monomeric Fc. We have also edited Fig.1a to annotate MFc3 as "MFc1-S354E" and MFc4 as "T1-S354E".

2) Do mFc3/mFc4 show binding to FcRn at physiological pH in a manner similar to YTE (on which they are partly based)? I think that would be important to show.

We observed similar (~50-fold) decreased levels in FcRn binding at neutral pH compared to pH 6. This has been added to the Results section on Page 10.

3) Report Vss?

We have added the modeled Vss values to Fig. 6B. In this report, our focus was on the bispecific molecules with slightly smaller sizes (100kDa) compared to IgG; as expected, we saw a small increase in Vss.

4) Finally can the authors comment on expression yields and also on stability to storage (stress test assays perhaps) as these features are of interest to readers interested in therapeutic applications of this technology?

We have added the expression yield for the bispecific molecules on Page 12 in the Results section. As we have so far observed reduced but reasonable DSF thermal stability of these variants (Fig. 2C), we hope to further characterize the developability and PK properties in non-human primates and have added a comment to this effect on Page 15 of the Discussion section.

Reviewer #2 (Remarks to the Author):

This manuscript describes novel mutations of monomeric Fc constructs intended to facilitate the crystallization of Fc-FcRn complexes and to be therapeutically applied. They present convincing data for the mutations identified that can be used in stable monomeric Fc constructs with tunable PK properties. These informations and this approach is useful for the Fc engineering community. The presentation of the work is solid, described in sufficient detail and I have no major comments or concerns

We appreciate the reviewer's positive remarks and recognition.

Reviewer #3 (Remarks to the Author):

Review Manuscript#: COMMSBIO-20-1189-T

Title: In vivo pharmacokinetic enhancement of monomeric Fc and monovalent bispecific designs through structural guidance

Major claims of the paper:

The authors of the Manuscript (COMMSBIO-20-1189-T) describe in a scientific interesting way the antibody scaffold engineering, necessary to design an FcRn binding active monomeric Fc construct.

The description of their structural guided approach to reach improvement in in vivo serum half-life is straightforward.

Starting from a previously known monomeric Fc variant, the work presented, describe additional analysis of the Fc dimer interface and the interplay of the FcRn binding CH3 area.

The authors have tested the influence of the residue S354 on thermostability, solved the crystal structure of two variants combining the new identified favorable S354E mutation with the previous known YTE mutation(MFc3) and a new N434Y & Y436L combination. (MFc4).

These variants have been tested by SEC-MALS & analytical ultracentrifugation, FcRn binding analysis utilizing Octet and in vivo PK in hFcRn transgenic mice.

This is a readable follow up of a previous research article from 2016, authored by the same group.

To publish in a Journal, addressing a broad audience, I am missing data describing the new behavior/biology behind a monomeric Fc such as the expected impact to the human immune

system

1)What is the difference in cellular trafficking of such constructs? Is the routing of a monomeric Fc, possessing improved FcRn binding affinities, similar to a weaker binding symmetric WT IgG1 or can it be found in other cellular compartments?

We appreciate the reviewer's question, as it suggests experiments to further improve the monomeric Fc platform. There has been evidence of differential trafficking through cellular compartments between monovalent vs. bivalent receptor engagement (for example, Niewoehner et al., 2014¹), with less lysosomal degradation and increased transcytosis via monovalent engagement. Although this is beyond the scope of the work described in our manuscript, this would be an interesting future work. We have incorporated a reflection on this matter in the Discussion section on Pages 15 and 16.

2)Is the pH dependent cellular recycling mechanism affected or not? Testing in a cellular FcRn assay would provide more insight on this aspect, ideally different FcRn expression levels and in presence of relevant human IgG levels. Determining recycling and transcytosis efficiency.

Thanks for these comments and suggestions. We have addressed these comments in the Discussion section. We observed pH dependence in the FcRn binding with these monomeric Fc variants (see Supplemental Figure S1). The increased FcRn binding at low pH resulted in improved cellular recycling, as indicated by the significantly improved protein half-life in human FcRn transgenic mice. We hope to validate the PK improvement in a cynomolgus model as a future plan.

3)Serum IgG competition. How does such a molecule behave in a more human like situation? (e.g. in the mouse model utilized in Lee CH, et al. Nat Commun. 2019. PMID: 31695028 or a IgG competition model (Stapleton NM, Nat Commun. 2011 Dec 20;2:599. doi: 10.1038/ncomms1608))

Thanks for these comments and suggestions. We have added in the Discussion a statement that we hope to validate the PK improvement in a cynomolgus model as a future plan.

4)How does the Fab and scFv of this monomeric Fc affect FcRn binding in contrast to a one armed dimeric Fc plus scFv construct?

We have found that the Fab and scFv attachments on the monomeric Fc maintained, within a two- to threefold difference, their FcRn binding with the Fc domain alone. This is mentioned on Page 12 of the Results section. We have not compared this with the Fab or scFv attachment of the one-armed dimeric Fc formats.

5)Is there an immunogenic impact expected of an engineered CH3 domain that is usually not accessible?

¹ Niewoehner J, Bohrmann B, Collin L, et al. Increased brain penetration and potency of a therapeutic antibody using a monovalent molecular shuttle. Neuron 2014;81(1):49–60.

We appreciate this valuable comment from the reviewer. We have added a discussion section on Page 15, as well as an in silico T-cell epitope prediction, to address this topic: “The exposure of the CH3 domain due to the disruption of the Fc dimer is not a new phenomenon. IgG4 backbone had been chosen for our MFc platform due to the ready engagement of Fab-arm exchange of the IgG4 molecules which alternates the Fc between a dimer-monomer equilibrium. In addition, to launch a strong and specific immune response, T-cell epitopes need to be processed and presented by antigen presenting cells. An in silico T cell epitope prediction around the monomer-forming mutations in MFc1 and MFc4 (Fig. S4) did not flag significant increase of HLA presentation risks for these mutations.”

6) A monomeric Fc is usually not expected to bind Fc gamma receptors, but is this true for high target expressing cells? An experiment that proves if a weak Fc gamma receptor binding is restored by high avidity on a highly Fab-mFc decorated target cell will be helpful to answer such a question. Especially of interest because the monomeric Fc is stabilized.

We appreciate this suggestion from the reviewer. We have followed up with experiments to indicate that the Fc gamma receptor engagement is indeed silenced in the monomeric Fc format. In contrast with onartuzumab, Fab fused to knob-and-hole Fc, the bivalent Fc counterpart to Onart-Fab-mFc, and Onart-Fab-MFc1 did not result in ADCC signals, even in higher-expressing cells. These results have now been added as Fig. S5 and are discussed on Page 16 of the Discussion section.

I would not recommend the publication of this manuscript in its current form; however, I would reconsider reviewing the manuscript after substantial addition of data addressing my questions above.

Kind regards Tilman Schlothauer

REVIEWERS' COMMENTS:

Reviewer #3 (Remarks to the Author):

Dear authors,

After reading the updated manuscript, I have the impression that my comments from the first review were not handled in the right context.

My question about the specific FcRn mediated trafficking as this is an undirected uptake to endothelial cells and following pH dependent binding to FcRn in the endosome to rescue antibodies or Fc-constructs from lysosomal degradation was answered with a different cellular uptake mechanism and other monovalent/bivalent related trafficking mechanisms that are away from FcRn mediated routing. Please update this to a more relevant citation.

The manuscript as it is describes nicely the structural aspects of the constructs and from this perspective reviewer 2 has no major comments and concerns

However looking to the biologic relevance, currently there are only human FcR transgenic mouse data included as it has been already described in Shan et al 2016

In my opinion, the manuscript would benefit from highlighting the FcRn binding properties in more human relevant assay setups.

As in the FcRn transgenic mouse model no other antibodies are present a monomeric Fc can be overinterpreted, therefore I wanted to suggest experiments that addresses the monomeric Fc behavior in presence of serum IgG.

Sure future cynomolgus PK studies would address this as the authors suggested in their reviewer response, but there are other experiments possible to bridge this gap.

In conclusion I am still not in favor of suggesting this manuscript for publication in Communications Biology.

Nevertheless, if the editors feel the authors have addressed enough of the other two reviewers' comments and you like to publish the manuscript this is fine with me